# Demographic, behavioural and occupational risk factors associated with SARS-CoV-2 infection in UK healthcare workers: a retrospective observational study

Daniel James Cooper [ORCID],[1,2] Sara Lear,[2] Nyarie Sithole,[2] Ashley Shaw,[3] Hannah Stark,[4] Mark Ferris [ORCID],[5] CITIID-NIHR BioResource COVID-19 collaboration consortium, John Bradley,[2,6] Patrick Maxwell,[1,2] Ian Goodfellow,[7] Michael P Weekes,[2,8] Shaun Seaman [ORCID],[9] Stephen Baker[10]

For numbered affiliations see end of article.

**Correspondence to**
Dr Daniel James Cooper;
dc801@cam.ac.uk

## ABSTRACT

**Objective** Healthcare workers (HCWs) are at higher risk of SARS-CoV-2 infection than the general population. This group is pivotal to healthcare system resilience during the COVID-19, and future, pandemics. We investigated demographic, social, behavioural and occupational risk factors for SARS-CoV-2 infection among HCWs.

**Design/setting/participants** HCWs enrolled in a large-scale sero-epidemiological study at a UK university teaching hospital were sent questionnaires spanning a 5-month period from March to July 2020. In a retrospective observational cohort study, univariate logistic regression was used to assess factors associated with SARS-CoV-2 infection. A Least Absolute Shrinkage Selection Operator regression model was used to identify variables to include in a multivariate logistic regression model.

**Results** Among 2258 HCWs, highest ORs associated with SARS-CoV-2 antibody seropositivity on multivariate analysis were having a household member previously testing positive for SARS-CoV-2 antibodies (OR 6.94 (95% CI 4.15 to 11.6); p<0.0001) and being of black ethnicity (6.21 (95% CI 2.69 to 14.3); p<0.0001). Occupational factors associated with a higher risk of seropositivity included working as a physiotherapist (OR 2.78 (95% CI 1.21 to 6.36); p=0.015) and working predominantly in acute medicine (OR 2.72 (95% CI 1.57 to 4.69); p<0.0001) or medical subspecialties (not including infectious diseases) (OR 2.33 (95% CI 1.4 to 3.88); p=0.001). Reporting that adequate personal protective equipment (PPE) was 'rarely' available had an OR of 2.83 (95% CI 1.29 to 6.25; p=0.01). Reporting attending a handover where social distancing was not possible had an OR of 1.39 (95% CI 1.02 to 1.9; p=0.038).

**Conclusions** The emergence of SARS-CoV-2 variants and potential vaccine escape continue to threaten stability of healthcare systems worldwide, and sustained vigilance against HCW infection remains a priority. Enhanced risk assessments should be considered for HCWs of black ethnicity, physiotherapists and those working in acute medicine or medical subspecialties. Workplace risk

## STRENGTHS AND LIMITATIONS OF THIS STUDY

⇒ A strength of this study was the use of a large, well-defined cohort of UK healthcare workers (HCWs).
⇒ The identification of actionable risk factors for mitigation of HCW infection.
⇒ Representative and transferable conclusions for acute hospital trusts.
⇒ Limitations include some potential retrospective recall bias of subjective questionnaire responses.

reduction measures include ongoing access to high-quality PPE and effective social distancing measures.

## BACKGROUND

The COVID-19 pandemic continues to overwhelm healthcare services globally with substantial morbidity and mortality.[1] The COVID-19 vaccination programme has been a major success in the UK, having a major impact on reducing hospitalisation and death.[2 3] However, the recent upsurge of cases associated with the delta variant[4] followed by emergence and dominance of the Omicron variant[4] illustrates how management of the pandemic requires sustained vigilance from the general public, policymakers and healthcare workers (HCWs). Notably, the delta[5] and omicron[6] variants have increased transmissibility, and a reduced efficacy of vaccination for prevention of infection.[7–10] Therefore, the emergence of additional variants with the potential for vaccine escape is a genuine concern for how we control SARS-CoV-2 in the long term.

HCWs are at a disproportionately high risk of infection from SARS-CoV-2[11] but remain key to the resilience of the health

service during this and all future pandemics. Infections of HCWs with SARS-CoV-2 and the isolation of contacts have resulted in significant staff shortages and increased strain on UK hospitals. Staff absence during September 2021 (most recent available figures) was 5.4% across the National Health Service (NHS), higher than August 2021 (5.1%) and higher than September 2020 (4.2%).[12] This high level of absence is despite the high rates of vaccination in HCWs, where up to 92.3% of staff in NHS trusts have received at least two doses of vaccine as of 28 February 2022.[13] Measures to reduce the risk of SARS-CoV-2 exposure to HCWs alongside widespread vaccination are vital to create resilience within the healthcare system. We have previously identified several occupational factors associated with increased risk of SARS-CoV-2 seropositivity in HCWs, which included job role, work location and ethnicity.[14] We conducted a retrospective observational cohort study in HCWs working in a major tertiary referral centre in the East of England with the objective of further elucidating the social, demographic, occupational and physical factors that may contribute to a higher risk of SARS-CoV-2 infection in HCWs.

## METHODS
### Population and setting
Cambridge University Hospitals NHS Foundation Trust (CUH) is a tertiary referral centre and teaching hospital with 1000 beds and 11 545 staff serving a population of 580 000 people in the East of England. The facility was equipped with 43 intensive care unit (ICU) beds prior to the pandemic, rising to 103 ICU beds at the peak of the pandemic, and an emergency department that receives ~14 000 attendees a month. During the study period (between March and June 2020), CUH treated 525 patients with PCR-confirmed COVID-19. The peak of COVID-19 admissions occurred in late March and early April 2020, with comparatively few COVID-19 admissions from June 2020 to November 2020. The definition of COVID-19 working for the purpose of risk stratification included clinical areas caring for patients with PCR-confirmed SARS-CoV-2 infection and those with patients for whom there is a high clinical suspicion of COVID-19, awaiting the results of SARS-CoV-2 PCR tests.

According to the 2011 England and Wales census,[15] 85.3% of the population of the East of England are white British, 5.5% are white other, 4.8% are Asian, 2% are black and 1.9% are of mixed ethnicity. The proportion of black, Asian and minority ethnic staff employed at CUH at the time of the study was largely representative of the overall NHS workforce[16] (21.2% vs 20.7%, respectively).

A staff screening programme for SARS-CoV-2 serological testing was available from 10 June 2020 to 7 August 2020 and has been described previously[14] (detailed in figure 1). In brief, all staff members were invited by email to participate in the serological screening programme and asked to self-refer for a clinic appointment. Written informed consent was obtained from all participants enrolled into this study. As part of this process, all

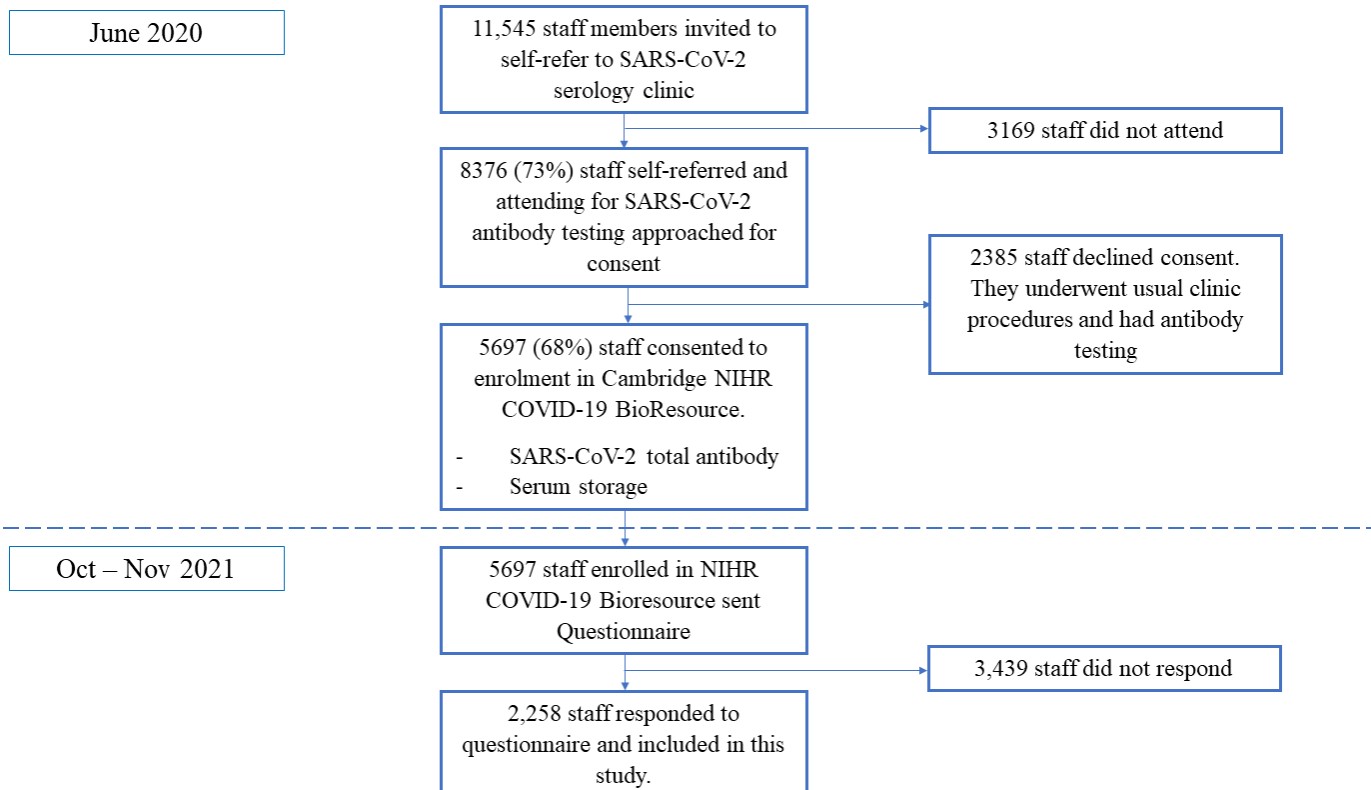

**Figure 1** Flow chart of study procedures. NIHR, National Institute for Health and Care Research.

participants were invited to join the National Institute for Health and Care Research BioResource–COVID-19 Research Cohort. Basic demographic and occupational information was recorded and a serum sample was taken and assayed for total SARS-CoV-2 antibodies (detailed below).

As no prior data were available to assess between-group differences on the metrics assessed in this study, a formal sample size calculation was not feasible.

## Questionnaire

A questionnaire covering demographic, occupational and behavioural factors potentially associated with risk of infection was designed with input and pretesting from infectious disease physicians, occupational physicians, virologists, microbiologists and epidemiologists. Formal reliability testing was not performed. Participants previously enrolled in a longitudinal HCW serological study (as described above) were invited by email to complete an online form containing the questionnaire in English, with the option to request the questionnaire in another language. A copy of the questions included in this questionnaire is included in the online supplemental appendix 1. Questionnaire invites were sent between October and November 2021 and questions within them related to participants' recalled behaviour during two periods: March–May 2020 and June–July 2020. Questions relating to behavioural and demographic factors were separated by time periods covering March–May and June–July to account for differences in behaviour and exposures outside of occupational environments due to the instigation (March 2020) and easing (June 2020) of the first UK national 'lockdown' measures.

## Laboratory assays

Serological testing for antibodies directed against SARS-CoV-2 was performed using the Centaur XP SARS-CoV-2 Total Antibody assay (Siemens Healthcare Limited, Surrey, UK). This method is a fully automated high throughput enzyme-linked chemiluminescent bridging immunoassay which targets the S1RBD antigen of SARS-CoV-2 and can detect all immunoglobulin subclasses (IgG, IgM and IgA). The method was independently validated by Public Health England and has a reported sensitivity and specificity of 98.1% (95% CI 96.6% to 99.1%) and 99.9% (95% CI 99.4% to 100%),[17] respectively. Samples were processed in the Biochemistry Laboratory at CUH following the standard operating procedure as stated by the manufacturer in their Instruction for Use after a local verification using guidance from the Royal College of Pathologists.[18]

## Statistical analysis

Univariate logistic regression was used to assess each variable in the questionnaire for association with positive SARS-CoV-2 antibodies. Variables with a p value of <0.05 on univariate analysis were included in a Least Absolute Shrinkage and Selection Operator (LASSO) regression analysis with post-estimation extended Bayesian information criterion commands for variable selection to include in a multivariate logistic regression model. The LASSO method of variable selection was used, in preference to the older stepwise selection method, because it has been shown to lead to higher prediction accuracy and variable selection that is less sensitive to small changes in the data.[19 20] Variables selected by LASSO analysis were included in a final multivariate logistic regression model. Data were analysed using Stata V.14.2 (StataCorp, College Station, Texas, USA).

## Patient and public involvement

Staff at CUH contributed to study and questionnaire design.

# RESULTS

## Baseline characteristics

A total of 2258 of 5698 (40%) invited HCWs responded to the invitation to complete an online questionnaire. Of the participants who responded to join the study, 19.65% (400 of 2044 responses) were male, the median age was 42 years (IQR 32–53 years) and 27.7% (618 of 2044) reported working in a designated COVID-19 'red' area during the first wave of the pandemic. Notably, 9.8% (n=222 of 2044) of the cohort tested seropositive for SARS-CoV-2 antibodies. The demographics of the study group are shown in table 1. Full details of variables and questionnaire responses are available in the online supplemental tables A–E.

## Univariate analysis

Responses demonstrated to have a significant association (p<0.05) with seropositivity in a univariate analysis are described in table 2 (the ORs and p values for responses to all questions are listed in online supplemental tables A–E). Noteworthy variables significantly associated with seropositivity for SARS-CoV-2 antibodies included having a household member who had tested positive for SARS-CoV-2 by PCR prior to staff serology testing (OR 3.48 (95% CI 2.09 to 5.78); p<0.001), or had tested positive by a SARS-CoV-2 antibody test (OR 11.3 (95% CI 7.08 to 18.01); p<0.001), or had had a household member who had been symptomatic (OR 3.71 (95% CI 2.8 to 4.96); p<0.001). Other demographic factors that were positively associated with seropositivity include identifying as being Asian or Asian British–other (OR 2.14 (95% CI 1.27 to 3.60); p=0.004), mixed ethnicity (OR 4.68 (95% CI 1.20 to 18.29); p=0.027), or black or black British–African ethnicity (5.74 (95% CI 2.61 to 12.60); p<0.001). Notably, reporting being born in the UK was associated with a protective effect (OR 0.59 (95% CI 2.8 to 4.96); p<0.001).

Renting a room in a shared house (OR 1.84 (95% CI 1.22 to 2.74); p=0.003) and living with another healthcare worker (OR 1.49 (95% CI 1.10 to 2.02); p=0.009) were further demographic factors associated with a significantly higher risk of infection on univariate analysis.

Table 1  Participant characteristics

| Baseline variable | n (%) |
|---|---|
| Sex (male) | 400/2044 (20) |
| Age (years), median (IQR) | 42 (32–53) |
| Ethnicity | |
| White British | 1584 (70) |
| White Irish | 35 (1.6) |
| White (other) | 294 (13) |
| Asian or Asian British (Indian) | 70 (3.1) |
| Asian or Asian British (Pakistani) | 8 (0.4) |
| Asian or Asian British (Bangladeshi) | 2 (0.1) |
| Asian or Asian British (other) | 116 (5.1) |
| Black or black British (Caribbean) | 7 (0.3) |
| Black or black British (African) | 29 (1.3) |
| Black or black British (other) | 1 (0.04) |
| Mixed—white and black Caribbean | 6 (0.3) |
| Mixed—white and black African | 9 (0.4) |
| Mixed—white and Asian | 12 (0.5) |
| Mixed—other | 10 (0.4) |
| Chinese | 27 (1.2) |
| Any other ethnic group | 26 (1.2) |
| Not stated | 19 (0.8) |
| Occupation | |
| Administrative staff | 336 (14.9) |
| Staff nurse | 298 (13.2) |
| Senior nursing staff | 311 (13.8) |
| Consultant | 150 (6.6) |
| Junior doctor | 88 (3.9) |
| Laboratory staff | 141 (6.3) |
| Healthcare assistant | 189 (8.4) |
| Theatre staff | 25 (1.1) |
| Manager | 118 (5.2) |
| Radiographer | 62 (2.8) |
| Midwife | 65 (2.9) |
| Physiotherapist | 36 (1.6) |
| Pharmacy staff | 51 (2.3) |
| Cleaning/domestic staff | 6 (0.3) |
| Dietitian | 23 (1) |
| Occupational therapist | 16 (0.7) |
| Speech and language therapist | 20 (0.9) |
| Porter | 7 (0.3) |
| Other | 314 (13.9) |
| COVID-19 working | 618/2235 (28) |

Other than job role, specialty and direct COVID-19 patient care, a number of other occupational factors were associated with higher odds of infection, including working night shifts (OR 1.68 (95% CI 1.26 to 2.25); p<0.001), using the doctors' mess (OR 1.77 (95% CI 1.17 to 2.69); p=0.007), spending rest or meal time with colleagues 'most of the time' (OR 1.99 (95% CI 1.19 to 3.33); p=0.009) and using hospital-supplied scrubs (OR 1.15 (95% CI 1.04 to 1.27); p=0.007). Those reporting having received formal personal protective equipment (PPE) training had a 40% higher risk of infection (OR 1.4 (95% CI 1.05 to 1.85); p=0.02) than those who did not. A higher proportion of those who worked in COVID-19 red areas reported receiving formal PPE training (486 of 613, 79%) than those not working in COVID-19 red areas (646 of 1594, 41%; p<0.0001), and formal PPE training no longer remained significant when controlling for 'red area' working (OR 1.2 (95% CI 0.88 to 1.63); p=0.20).

Those reporting having adequate PPE available 'some of the time' (OR 1.93 (95% CI 1.22 to 3.05); p=0.005) or 'rarely' (OR 3.60 (95% CI 1.71 to 7.57); p=0.001) were associated with higher odds of infection compared with those who reported adequate PPE being available 'all of the time'. A higher proportion of those reporting PPE being available 'some of the time' (78 of 194, 40%) worked in COVID-19 red areas compared with those reporting PPE being available 'all of the time' (540 of 2041, 27%; p<0.0001). Attending shift handover (a staff meeting prior to shift change) where social distancing was not possible was associated with a higher risk of infection (OR 1.74 (95% CI 1.31 to 2.30); p<0.001). Working predominantly from home between March and June 2020 was associated with a protective effect (OR 0.60 (95% CI 0.39 to 0.91); p=0.016), as was working from home between June and July 2019 (OR 0.58 (95% CI 0.36 to 0.94); p=0.026).

Reporting being a smoker was associated with a lower risk of infection (OR 0.37 (95% CI 0.18 to 0.76); p=0.007) among behavioural risk factors. Reporting drinking alcohol was associated with a lower risk of infection (OR 0.74 (95% CI 0.55 to 0.98); p=0.38); however, frequency of drinking alcohol had no effect on risk of infection. Having food or grocery deliveries to home 'daily' was associated with a higher risk of infection between March and May 2020 (OR 5.38 (95% CI 1.27 to 22.8); p=0.022) and June and July 2020 (OR 6.1 (95% CI 1.01 to 36.7); p=0.049) compared with those who reported 'never' having food or groceries delivered. Exercising outdoors 'daily' was associated with a lower risk of infection between March and May 2020 (OR 0.58 (95% CI 0.39 to 0.86); p=0.007) and between June and July 2020 (OR 0.56 (95% CI 0.37 to 0.84); p=0.005) compared with those who reported exercising outdoors less than once per week.

### LASSO model fitting
The variables selected by the LASSO model are shown in table 3, and included having a household member who had tested positive for SARS-CoV-2 antibodies or had had a positive SARS-CoV-2 PCR test, a household member previously displaying symptoms synonymous with COVID-19, black ethnicity, working as a physiotherapist, reporting working in acute medicine or

**Table 2** Significant univariate analysis variables

| | Variable | OR* | 95% CI | P value | n (positive)/N (responses) (%) |
|---|---|---|---|---|---|
| Demographic | | | | | |
| | Rent room in shared house | 1.84 | 1.22 to 2.74 | 0.003 | 33/209 (16) |
| | Live with other HCWs | 1.49 | 1.10 to 2.02 | 0.009 | 70/550 (13) |
| | Children attended school in June | 0.58 | 0.35 to 0.97 | 0.038 | 30/395 (7.6) |
| | Household member +ve PCR test | 3.48 | 2.09 to 5.78 | <0.0001 | 22/84 (26) |
| | Household member +ve Ab test | 11.29 | 7.08 to 18.01 | <0.0001 | 40/79 (51) |
| | Household member symptomatic | 3.71 | 2.8 to 4.96 | <0.0001 | 95/437 (22) |
| | Born in UK | 0.59 | 0.44 to 0.79 | <0.001 | 136/1616 (8.4) |
| | Ethnicity | 1.06 | 1.03 to 1.10 | <0.001† | 1584/2258‡ |
| Occupational | | | | | |
| | Job role | | | | |
| | Admin staff | 1 | — | — | 24/336 (7) |
| | Staff nurse | 2.02 | 1.18 to 3.43 | 0.01 | 40/298 (13.4) |
| | Physiotherapist | 4.33 | 1.83 to 10.25 | 0.001 | 9/36 (25) |
| | Direct patient care COVID-19 | 1.86 | 1.41 to 2.47 | <0.001 | 103/757 (13.6) |
| | Worked in red area | 1.78 | 1.33 to 2.38 | <0.001 | 85/618 (13.8) |
| | Specialty | | | | |
| | Non-patient-facing roles | 1 | — | — | 10/169 (6) |
| | Critical care | 2.51 | 1.10 to 5.77 | 0.029 | 16/117 (13.7) |
| | Acute medicine | 4.57 | 2.08 to 10.07 | <0.001 | 23/103 (22.3) |
| | Medical specialties | 4.35 | 2.01 to 9.42 | <0.001 | 26/121 (21.5) |
| | Surgical | 2.71 | 1.24 to 5.93 | 0.012 | 22/151 (14.6) |
| | Night shifts | 1.68 | 1.26 to 2.25 | <0.001 | 82/604 (13.6) |
| | Receive formal PPE training | 1.40 | 1.05 to 1.85 | 0.02 | 129/1141 (11.3) |
| | Adequate PPE available | | | | |
| | All of the time | 1 | — | — | 83/1038 (8) |
| | Most of the time | 1.34 | 0.98 to 1.83 | 0.065 | 92/882 (10.4) |
| | Some of the time | 1.93 | 1.22 to 3.05 | 0.005 | 28/195 (14.4) |
| | Rarely | 3.60 | 1.71 to 7.57 | 0.001 | 10/42 (23.8) |
| | Rest/meal with colleagues | | | | |
| | Never | 1 | — | — | |
| | Most of the time | 1.99 | 1.19 to 3.33 | 0.009 | § |
| | Use doctors' mess | 1.77 | 1.17 to 2.69 | 0.007 | 30/195 (15.4) |
| | Hospital-supplied scrubs | 1.15 | 1.04 to 1.27 | 0.007 | |
| | Work from home—March | 0.60 | 0.39 to 0.91 | 0.016 | |
| | Work from home—June | 0.58 | 0.36 to 0.94 | 0.026 | |
| | Handover w/o social distancing | 1.74 | 1.31 to 2.30 | <0.0001 | |
| Behavioural | | | | | |
| | Smoker | 0.37 | 0.18 to 0.76 | 0.007 | |
| | Food deliveries—March | | | | |
| | Daily | 5.38 | 1.27 to 22.8 | 0.022 | ¶ |
| | Food deliveries—June | | | | |
| | Daily | 6.10 | 1.01 to 36.7 | 0.049 | ¶ |
| | Exercise outdoors—March | | | | |
| | Daily | 0.58 | 0.39 to 0.86 | 0.007 | ¶ |

Continued

**Table 2** Continued

| Variable | OR* | 95% CI | P value | n (positive)/N (responses) (%) |
|---|---|---|---|---|
| Exercise outdoors—June | | | | |
| Daily | 0.56 | 0.37 to 0.84 | 0.005 | ¶ |

*Unadjusted OR.
†P value for likelihood ratio test.
‡Number of participants identifying as white British.
§Compared with 'never'.
¶Compared with <once per week.
Ab, antibody; HCWs, healthcare workers; PPE, personal protective equipment.

medical subspecialties, reporting that adequate PPE was 'rarely' available, working in a designated 'red' area and attending handovers where adequate social distancing was not possible.

## Multivariate analysis

We used a multivariate logistic regression model to include all variables selected by LASSO modelling. In this model, working in a designated COVID-19 area and having a household member with a previous positive SARS-CoV-2 PCR swab were not significantly associated with the participant having a positive antibody test result (p>0.05) and were dropped from the final model. A total of eight variables were included in the final multivariate logistic regression model (table 3 and figure 2).

In this resulting model, the highest reported adjusted ORs (aORs) associated with participants testing seropositive for SARS-CoV-2 antibodies were having a household member who had previously tested positive for SARS-CoV-2 antibodies (OR 6.94 (95% CI 4.15 to 11.6); p<0.0001) and being of black ethnicity (6.21 (95% CI 2.69 to 14.3); p<0.0001). Occupational factors associated with a higher risk of seropositivity were working as a physiotherapist (aOR 2.78 (95% CI 1.21 to 6.36); p=0.015) and reporting that they predominantly worked in acute medicine (aOR 2.72 (95% CI 1.57 to 4.69); p<0.0001) or medical subspecialties (not including infectious diseases) (aOR 2.33 (95% CI 1.4 to 3.88); p=0.001). Reporting that adequate PPE was 'rarely' available was associated with an aOR of 2.83 (95% CI 1.29 to 6.25; p=0.01) and reporting

attending a handover where social distancing was not possible was associated with an aOR of 1.39 (95% CI 1.02 to 1.9; p=0.038).

## DISCUSSION

In this systematic evaluation of demographic, occupational and behavioural risk factors associated with COVID-19 seropositivity among HCWs, we have identified several targetable risk factors for HCW infection from SARS-CoV-2. These may also serve as a framework for targeting HCW risk during future respiratory pathogen pandemics. The ability of healthcare systems to cope with the surge of infections requiring hospitalisation has been challenged in a number of countries including the UK,[21] India,[22] USA[23] and Brazil[24] and resulted in excess deaths.[23 25] The resilience of a healthcare system relies heavily on staff remaining well and able to work. HCWs have been disproportionately affected by infection rates[26 27] during this pandemic.

Both a positive SARS-CoV-2 antibody in a household member and prior symptoms in a household member were significantly associated with seropositivity in a multivariate model. The finding that a positive PCR test in a household member was not associated with seropositivity on multivariate analysis may reflect a proportional relationship between viral load and transmissibility in asymptomatic infections. A study of cycle threshold (Ct) values (as a proxy for viral load) in uncomplicated community SARS-CoV-2 demonstrated that self-reported symptoms were an independent predictor of lower Ct value (ie, higher viral load), and that Ct values were significantly higher in those who remained antibody negative.[28] Taken together, these results suggest that a household member with positive symptoms (and either untested or false negative test) or a high enough viral load to develop antibodies contributes more to risk of infection in household members than a positive PCR test alone.

The finding that black ethnicity remained highly significantly associated with seropositivity after controlling for many plausible explanations is concerning. The effect of increased risk of infection in certain ethnicities has been reported elsewhere; the reasons for this are complex and remain poorly understood but may include increased risk of household transmission.[29] South Asian and black

**Table 3** Final multivariate model

| Variable | OR | 95% CI | P value |
|---|---|---|---|
| Household positive antibody | 6.94 | 4.15 to 11.6 | <0.001 |
| Household positive symptoms | 2.95 | 2.13 to 4.08 | <0.001 |
| Black ethnicity | 6.21 | 2.69 to 14.3 | <0.001 |
| Physiotherapist | 2.78 | 1.21 to 6.39 | 0.015 |
| Acute medicine specialty | 2.72 | 1.57 to 4.69 | <0.001 |
| Medical specialties | 2.33 | 1.40 to 3.88 | 0.001 |
| Inadequate PPE | 2.84 | 1.29 to 6.25 | 0.010 |
| Handover w/o distancing | 1.39 | 1.02 to 1.90 | 0.038 |

PPE, personal protective equipment.

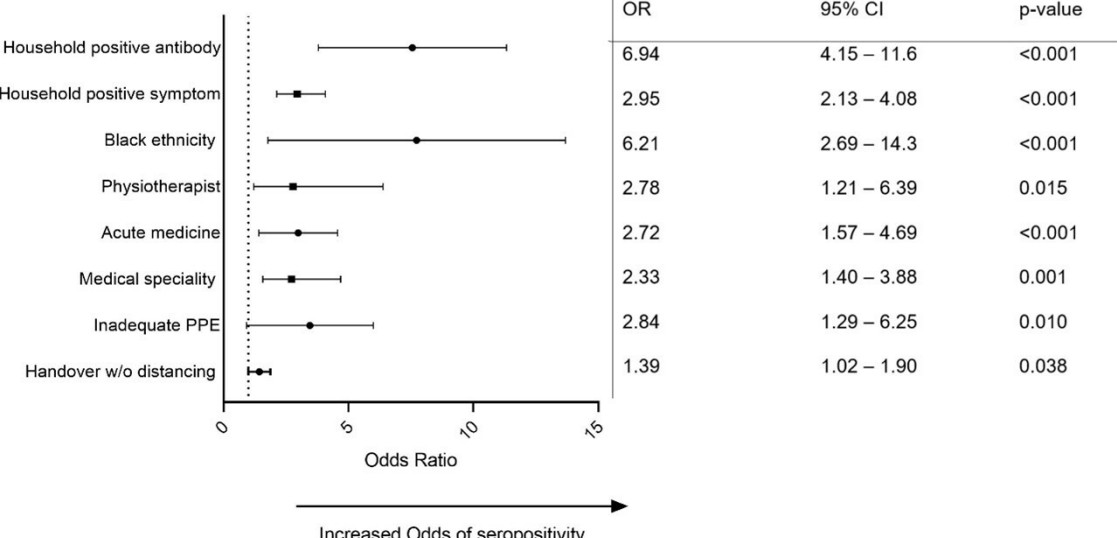

| | OR | 95% CI | p-value |
|---|---|---|---|
| Household positive antibody | 6.94 | 4.15 – 11.6 | <0.001 |
| Household positive symptom | 2.95 | 2.13 – 4.08 | <0.001 |
| Black ethnicity | 6.21 | 2.69 – 14.3 | <0.001 |
| Physiotherapist | 2.78 | 1.21 – 6.39 | 0.015 |
| Acute medicine | 2.72 | 1.57 – 4.69 | <0.001 |
| Medical speciality | 2.33 | 1.40 – 3.88 | 0.001 |
| Inadequate PPE | 2.84 | 1.29 – 6.25 | 0.010 |
| Handover w/o distancing | 1.39 | 1.02 – 1.90 | 0.038 |

**Figure 2** Forest plot of final multivariate model. PPE, personal protective equipment.

ethnicity have been found to be associated with a higher risk of hospitalisation, ICU admission and death relative to white ethnicity.[30] An increased risk of infection in non-white ethnicity has been reported across multiple other studies in other countries and healthcare settings, including black and Asian staff in UK hospitals,[31] black staff in US healthcare systems,[32 33] non-white workers in Brazil,[34] and black or Hispanic ethnicity in Canada.[35] Observational studies in countries not assessing ethnicity in HCW risk-factor analyses have reported risks that have been suggested as potentially contributing to health disparities in non-white ethnicities including income level, educational background and use of mass transit systems.[34 36]

The subjective feeling that adequate PPE was rarely available remained highly statistically significant in the final multivariate model. While interesting, this finding requires a careful consideration of context and the subjective nature of the question. The availability and standard of PPE at CUH have been reported as exceeding that recommended by Public Health England for HCWs during the period of the study.[37] Furthermore, we have demonstrated elsewhere that the use of this enhanced PPE was effective at reducing the risk of infection among HCWs.[37] CUH reported the second lowest number of hospital-acquired COVID-19 cases in the East of England[38] out of 14 hospital trusts (suggesting high standards of infection control), with clinical outcomes for patients with COVID-19 exceeding the national standard.[39] Despite these factors, 11% of staff reported the perception that adequate PPE was available 'some of the time' or 'rarely'. Similar data are not available for comparison at other NHS sites. Staff at a higher risk of occupational exposure to infectious patients are likely to have experienced higher rates of anxiety related to PPE and therefore recall that anxiety, especially within the wider context of the media reporting of the national and global effects of the COVID-19 pandemic during that time. This is demonstrated in the higher proportion of those reporting insufficient PPE being available 'some of the time' or 'rarely' working in COVID-19 red areas compared with those reporting adequate PPE being available 'all of the time'. Nevertheless, the fact that this variable remained highly significant after LASSO variable selection and inclusion in the multivariate model highlights the need for availability of effective PPE for all HCWs at occupational risk of infection. Effective PPE is key for reducing infection, as well as ensuring staff's mental well-being and reducing potential burnout.[40]

The impact of social distancing on the risk of COVID-19 infection is now well documented.[41 42] Our analysis suggests that the practice of social distancing and mask wearing during shift change handovers and other meeting times should continue to be encouraged as a modifiable behaviour that has the potential to decrease the risk of SARS-CoV-2 infection in HCWs.

Physiotherapy played a key role in both ICU and acute medical wards with therapeutic positioning, early mobilisation and breathing exercises.[43] In addition, the risk of hospitalisation with COVID-19 increases with age, and elderly populations constituted a large proportion of non-ICU hospital admissions.[44] Physiotherapists constitute an integral part of a face-to-face multidisciplinary team during acute hospital admissions for elderly people,[45] and would therefore have had significant exposure to SARS-CoV-2-infected patients. The increased risk of infection among physiotherapists during these activities requires further investigation and should be considered when assessing clinical practice risk and PPE standards.

These analyses have limitations. By their nature, questions about behavioural factors contain subjective answers and must be interpreted with caution, including the subjective experience of availability of PPE. In addition, the questionnaire was sent to participants 3–7 months following the period encompassed by the questions, which could add imprecision. This delay leaves

responses open to recall bias;however, most important factors assessed here (ethnicity, job role, prior household PCR and antibody results) are objective and are unlikely to have changed in the intervening period. Participants were aware of their serostatus at the time of completing the questionnaire, which may also have influenced responses to subjective questions, particularly around the availability of PPE. We have previously shown that porters and domestic staff are at a higher risk of infection[46]; however, their experience was not captured in this study due to low numbers of respondents (n=7 and n=0, respectively). These analyses cover the time period where the original wild-type Wuhan strain was the predominant circulating variant in the UK. Data on established and emerging variants, including the delta variant[4] and the now predominant Omicron variant,[6] suggest they may be more infectious and thus levels of risk and risk factors may not be identical. We think that the risk factors discussed within this paper are unlikely to be greatly affected by a change in the risk of infection in new variants and remain broadly generalisable as risk factors for HCW infection, although the widespread introduction of both population and HCW vaccination since this study is likely to have had a significant impact on these risk factors.[47]

Our work identified a number of targetable risk factors for mitigation of the risk of HCW infection during the ongoing COVID-19 pandemic. Maintaining vigilance and providing adequate social distancing space for shift change handover are likely to reduce the risk of HCW infection. The subjective experience of staff towards PPE should be considered when providing adequate and safe PPE provision and training. In addition, there are a number of non-modifiable risk factors, which nevertheless are feasible for extra mitigation strategies for healthcare professionals working within a health service to reduce the risk of HCW infection.

**Author affiliations**
[1]Department of Medicine, University of Cambridge School of Clinical Medicine, Cambridge, UK
[2]Cambridge University Hospital, Cambridge University Hospitals NHS Foundation Trust, Cambridge, UK
[3]Medical Director's Office, Cambridge University Hospitals NHS Foundation Trust, Cambridge, UK
[4]NIHR Bioresource, NIHR Cambridge Biomedical Research Centre, Cambridge, UK
[5]Occupational Health, Cambridge University Hospitals NHS Foundation Trust, Cambridge, UK
[6]Department of Medicine, University of Cambridge, Cambridge, UK
[7]Department of Pathology, Division of Virology, University of Cambridge, Cambridge, UK
[8]Cambridge Institute for Medical Research, University of Cambridge, Cambridge, UK
[9]MRC Biostatistics Unit, University of Cambridge, Cambridge, UK
[10]Cambridge Institute of Therapeutic Immunology and Infectious Disease, University of Cambridge School of Clinical Medicine, Cambridge, UK

**Acknowledgements** We thank National Institute for Health Research (NIHR) BioResource volunteers for their participation, and gratefully acknowledge NIHR BioResource centres, NHS trusts and staff for their contribution. We thank the NIHR, NHS Blood and Transplant, and Health Data Research UK as part of the Digital Innovation Hub Programme.

**Contributors** DJC, SL and SB conceived and designed the study. DJC, SB and SS conducted the analysis. DJC, SL, SB, NS, AS, HS, MF, PM, JB, MPW and IG contributed to questionnaire design and analysis. Operational input and analysis were provided by AS and MF. Study logistics, questionnaire distribution and data collection were performed by Cambridge NIHR BioResource and the CITIID-NIHR BioResource COVID-19 collaboration consortium, overseen by HS. All authors read the manuscript and provided edits. DJC accepts full responsibility for the work and/or the conduct of the study, had access to the data, and controlled the decision to publish.

**Funding** DJC and SL received funding for this work from Addenbrooke's Charitable Trust (grant ID 900254). The work was also funded by awards from NIHR to the NIHR BioResource (RG94028 & RG85445). This research was funded in part by the Wellcome Trust (215515/Z/19/Z Senior Fellowship to SB, 412 207498/Z/17/Z Senior Fellowship to IG, 108070/Z/15/Z Senior Fellowhsip to MPW). For the purpose of open access, the author has applied a CC BY public copyright licence to any Author Accepted Manuscript version arising from this submission.

**Disclaimer** The views expressed are those of the author(s) and not necessarily those of the NHS, the NIHR or the Department of Health and Social Care.

**Competing interests** None declared.

**Patient and public involvement** Patients and/or the public were involved in the design, or conduct, or reporting, or dissemination plans of this research. Refer to the Methods section for further details.

**Patient consent for publication** Not required.

**Ethics approval** This study involves human participants and ethical approval for this study was granted by the East of England–Cambridge Central Research Ethics Committee (IRAS ID: 220277). Participants gave informed consent to participate in the study before taking part.

**Provenance and peer review** Not commissioned; externally peer reviewed.

**Data availability statement** Data are available upon reasonable request.

**ORCID iDs**
Daniel James Cooper http://orcid.org/0000-0002-3643-7605
Mark Ferris http://orcid.org/0000-0001-5040-4263
Shaun Seaman http://orcid.org/0000-0003-3726-5937

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
