## [Reviewer comments · BMJ Open]

ARTICLE DETAILS

TITLE (PROVISIONAL)	A retrospective observational study of demographic, behavioural and occupational risk factors associated with SARS-COV-2 infection in UK healthcare workers
AUTHORS	Cooper, Daniel; Lear, Sara; Sithole, Nyarie; Shaw, Ashley; Stark, Hannah; Ferris, Mark; COVID-19 collaboration, CITIID-NIHR BioResource; Bradley, John; Maxwell, Patrick; Goodfellow, Ian; Weekes, Michael; Seaman, Shaun; Baker, Stephen

VERSION 1 – REVIEW

REVIEWER	Chang, Chee-Tao Hospital Raja Permaisuri Bainun, Clinical Research Centre
REVIEW RETURNED	06-May-2022

GENERAL COMMENTS	This paper described the risk factors of COVID-19 infection among healthcare workers. While similar findings were reported in available literatures, this paper may add consolidate further on risk factors among HCW in the local context. Overall, the paper is methodologically sound, but the discussion needs further depth and support with reasoning. Abstract-design: the study design was not clearly stated Abstract-conclusion: I am not sure whether the term “novel” is still relevant. Methods-Population and setting: This sentence was not clear and need rephrasing. “The definition of COVID-19 working for the purpose of risk stratification included clinical areas designated as either “Red” (patients with PCR-confirmed SARS-CoV-2 infection) or “Amber” (patients for whom there is a high clinical suspicion of COVID-19).” Methods-Questionnaire: there was a lack of description on the validation process of the questionnaire, is there any pre-testing and reliability testing performed? And whether the questionnaire was only available in one language? Methods- there was a lack of description on the sample size calculation. Methods- statistical analysis: It was not mentioned what software was used for statistical analysis Results: It is valuable to report the response rate, i.e. what is the total number of HCW invited? Results: I am not sure why OR was not included in this sentence. “Other demographic factors that were positively associated with seropositivity include identifying as being Asian or Asian British
--

	(other), mixed ethnicity, or Black or Black British (African) ethnicity” Discussion: Overall, there is a lack of comparison with similar studies from other countries, whether the risk factors for COVID-19 infection among HCW are the same or different, and the reason of why is it so? Discussion: first paragraph-authors mentioned that they have identified targetable risk factors for future pandemics. I believe that this is an over-assumption, because the study was based solely on this COVID-19 pandemic. Discussion-3rd paragraph: I believe there is still a lot of room for discussion regarding ethnicity. Rather than unknown, authors may discuss on health equalities, access to healthcare, housing, discrimination and many other factors regarding why minorities has higher risk of getting COVID-19 infections. One such example could be found at CDC: https://www.cdc.gov/coronavirus/2019-ncov/community/health-equity/race-ethnicity.html Discussion-4th paragraph: The last few sentences of the paragraph regarding social distancing comes abruptly after a long elaboration on PPE. The authors might consider expand the discussion regarding social distancing and putting it into a separate paragraph for better readability. Discussion-5th paragraph: I am not sure why “hospitalization with COVID-19 increase with age” comes into picture, because it was not presented in the Results section. It also did not cohere with the subsequent description on physiotherapist’s risk. Discussion-6th paragraph: “We think that the risk factors discussed within this paper are unlikely to be greatly affected by a change in the risk of infection in new variants and remain broadly generalisable as risk factors for HCW infection.” Authors may explain what is their reasoning for such assumption.
--	---

REVIEWER	Martineau, Adrian Queen Mary University of London
REVIEW RETURNED	12-Jul-2022

GENERAL COMMENTS	This well-written manuscript reports findings of a retrospective observational study investigating risks for anti-S IgG/A/M seropositivity in a cohort of 2258 UK healthcare workers. Strengths include the large size and detailed characterization of risks inside and outside the workplace, allowing for adjustment for potential confounders. The findings demonstrate the combined importance of exposure at home and in the workplace, esp. outside of ICU / Infectious Diseases wards where use/quality of PPE may be less/lower. The study also highlights a massive ethnic disparity in risk, which is currently unexplained. The serology test employed utilized detected 3 classes of antibody and has been validated with high sensitivity and specificity. There are some limitations though.  1. Domestic staff (n=0) and porters (n=7) were seriously under-represented – this is a limitation and should be explicitly acknowledged. These groups were potentially at very high risk of infection, but their experience has not been captured in this study. 2. The study was conducted in the pre-vaccination era, which could constrain relevance of its results for the current situation where the
---

	majority of HCWs are vaccinated against SARS-CoV-2. Population-based studies e.g. https://doi.org/10.1101/2022.03.11.22272276 have reported that vaccination has been effective in ablating / attenuating increased risk of SARS-CoV-2 infection associated with Black or Asian ethnic origin and increased BMI, for example. It was also conducted before emergence of delta / omicron variants, which could also constrain relevance for the current situation, but to a lesser extent. These limitations should be acknowledged in the Discussion. The Discussion currently states: “We think that the risk factors discussed within this paper are unlikely to be greatly affected by a change in the risk of infection in new variants and remain broadly generalisable as risk factors for HCW infection”. I tend to agree re variants – but the rollout of vaccination is likely a game changer that could dramatically affect risk factors reported here. 3. The authors acknowledge that the retrospective nature of the questionnaires could have introduced recall bias - especially likely to operate re subjective factors like PPE availability. Since respondents knew their serostatus at the time of questionnaire completion, this could have become a self-fulfilling prophecy (people who knew they got infected likely to blame lack of PPE). There is also a potential risk of imprecision in answers due to the time elapsed which should also be acknowledged. 4. Clarity re dates: I found the chronology difficult to understand. The authors write: “Questionnaire invites were sent between October and November 2020 and covered the period between March 2020 and June 2020 (the time of serological sampling). Questions relating to behavioural and demographic factors were separated by time periods covering March – May and June – July to account for differences in behaviour and exposures outside of occupational environments due to the instigation (March 2020) and easing (June 2021) of the first UK national “lockdown” measures’. Is there a typo here (June 2020?). ‘Covered the period’ is ambiguous, as it could refer to when the questionnaires were completed, or the time period referred to in the questionnaire. Suggest something along the lines of ‘Questionnaire invites were sent between October and November 2020 and questions within them related to participants’ recalled behaviour during two periods - March-May 2020 and June-July 2020’. Was the time of serological sampling in June 2020 only or was it done over 4 months (Mar 20 to Jun 20?). The wording is ambiguous. Elsewhere it is written that “HCWs were sent questionnaires spanning Mar-Aug 2021.’ I am struggling to reconcile this with other dates reported. A flow chart to illustrate the chronology of serology sampling, consent, questionnaire issue / completion etc would help to clarify the methods. 5. I was interested to see that BMI did not associate with risk of seropositivity – a contrast to findings from others e.g. PMID 35189888. Could the authors speculate why this may have been the case? 6. Table 1 (Participant characteristics) details age, sex, ethnicity and ‘COVID working’ – but there are lot more characteristics of interest here – job role, other exposures etc. It would be useful to flesh this out, esp with details of how different job roles were distributed. 7. What was the basis for the sample size? Was a power calculation done? Minor 1. Typos:
--	--

	Table C (p46/59): Pharmacy staff 7.51 – should this be 7/51? environments due to the instigation (March 2020) and easing (June 2021) of the first UK national “lockdown” measures’ – should this be June 2020? Elsewhere it is written that “HCWs were sent questionnaires spanning Mar-Aug 2021.’ Should this be March to July 2020? 2. Abstract and elsewhere – if odds ratios adjusted, then they should be reported as such
--	--

VERSION 1 – AUTHOR RESPONSE

Reviewer: 1

Comments to the Author:

This paper described the risk factors of COVID-19 infection among healthcare workers. While similar findings were reported in available literatures, this paper may add consolidate further on risk factors among HCW in the local context. Overall, the paper is methodologically sound, but the discussion needs further depth and support with reasoning.

Abstract-design: the study design was not clearly stated

This has been updated.

Abstract-conclusion: I am not sure whether the term “novel” is still relevant.

The wording has been changed.

Methods-Population and setting: This sentence was not clear and need rephrasing. “The definition of COVID-19 working for the purpose of risk stratification included clinical areas designated as either “Red” (patients with PCR-confirmed SARS-CoV-2 infection) or “Amber” (patients for whom there is a high clinical suspicion of COVID-19).”

This been updated to improve clarity

Methods-Questionnaire: there was a lack of description on the validation process of the questionnaire, is there any pre-testing and reliability testing performed? And whether the questionnaire was only available in one language?

This has been clarified.

Methods- there was a lack of description on the sample size calculation.

The was no indication for a sample size calculation, as this questionnaire was sent to all available respondents enrolled in the original longitudinal study. Additionally, the were no prior data available to assess differences between groups on the metrics assessed in the analysis to inform a sample size calculation.

This has been clarified in the text.

Methods- statistical analysis: It was not mentioned what software was used for statistical analysis

This has been included.

Results: It is valuable to report the response rate, i.e. what is the total number of HCW invited?

This has been included in the results text.

Results: I am not sure why OR was not included in this sentence. "Other demographic factors that were positively associated with seropositivity include identifying as being Asian or Asian British (other), mixed ethnicity, or Black or Black British (African) ethnicity"

This has been included.

Discussion: Overall, there is a lack of comparison with similar studies from other countries, whether the risk factors for COVID-19 infection among HCW are the same or different, and the reason of why is it so?

This has been updated.

Discussion: first paragraph-authors mentioned that they have identified targetable risk factors for future pandemics. I believe that this is an over-assumption, because the study was based solely on this COVID-19 pandemic.

We believe that these targetable risk factors could serve as a framework for targeting and lowering risk in HCWs during future pandemics, particularly those driven by respiratory pathogens. The wording of this sentence has been changed to clarify that this is not a firm conclusion.

Discussion-3rd paragraph: I believe there is still a lot of room for discussion regarding ethnicity. Rather than unknown, authors may discuss on health equalities, access to healthcare, housing, discrimination and many other factors regarding why minorities has higher risk of getting COVID-19 infections. One such example could be found at CDC: <https://www.cdc.gov/coronavirus/2019-ncov/community/health-equity/race-ethnicity.html>

This has been included.

Discussion-4th paragraph: The last few sentences of the paragraph regarding social distancing comes abruptly after a long elaboration on PPE. The authors might consider expand the discussion regarding social distancing and putting it into a separate paragraph for better readability.

This has been done.

Discussion-5th paragraph: I am not sure why "hospitalization with COVID-19 increase with age" comes into picture, because it was not presented in the Results section. It also did not cohere with the subsequent description on physiotherapist's risk.

This is in relation to the proportion of workload of physiotherapists being undertaken with elderly hospitalised patients. This paragraph has been restructured and clarified to clarify this.

Discussion-6th paragraph: "We think that the risk factors discussed within this paper are unlikely to be greatly affected by a change in the risk of infection in new variants and remain broadly generalisable as risk factors for HCW infection." Authors may explain what is their reasoning for such assumption.

This sentence was qualified by the preceding sentence stating that subsequent variant risk factors are not identical. Further, we have expanded to add that vaccination is likely to have had more impact on risk than subsequent variants per reviewer 2's comments below.

Reviewer: 2

Comments to the Author:

This well-written manuscript reports findings of a retrospective observational study investigating risks for anti-S IgG/A/M seropositivity in a cohort of 2258 UK healthcare workers. Strengths include the large size and detailed characterization of risks inside and outside the workplace, allowing for adjustment for potential confounders. The findings demonstrate the combined importance of exposure at home and in the workplace, esp. outside of ICU / Infectious Diseases wards where use/quality of PPE may be less/lower. The study also highlights a massive ethnic disparity in risk, which is currently unexplained. The serology test employed utilized detected 3 classes of antibody and has been validated with high sensitivity and specificity.

There are some limitations though.

1. Domestic staff (n=0) and porters (n=7) were seriously under-represented – this is a limitation and should be explicitly acknowledged. These groups were potentially at very high risk of infection, but their experience has not been captured in this study.

This has been explicitly acknowledged – and referenced to our previous work, where porters and domestic staff were at higher risk of infection.

2. The study was conducted in the pre-vaccination era, which could constrain relevance of its results for the current situation where the majority of HCWs are vaccinated against SARS-CoV-2. Population-based studies e.g. <https://doi.org/10.1101/2022.03.11.22272276> have reported that vaccination has been effective in ablating / attenuating increased risk of SARS-CoV-2 infection associated with Black or Asian ethnic origin and increased BMI, for example. It was also conducted before emergence of delta / omicron variants, which could also constrain relevance for the current situation, but to a lesser extent. These limitations should be acknowledged in the Discussion.

The Discussion currently states: “We think that the risk factors discussed within this paper are unlikely to be greatly affected by a change in the risk of infection in new variants and remain broadly generalisable as risk factors for HCW infection”. I tend to agree re variants – but the rollout of vaccination is likely a game changer that could dramatically affect risk factors reported here.

Discussions around these suggestions have been included and expanded.

3. The authors acknowledge that the retrospective nature of the questionnaires could have introduced recall bias - especially likely to operate re subjective factors like PPE availability. Since respondents knew their serostatus at the time of questionnaire completion, this could have become a self-fulfilling prophecy (people who knew they got infected likely to blame lack of PPE). There is also a potential risk of imprecision in answers due to the time elapsed which should also be acknowledged.

Both of these points are already acknowledged and discussed in the existing limitations paragraph, however have now been further expanded.

4. Clarity re dates: I found the chronology difficult to understand. The authors write:

“Questionnaire invites were sent between October and November 2020 and covered the period between March 2020 and June 2020 (the time of serological sampling). Questions relating to behavioural and demographic factors were separated by time periods covering March – May and June – July to account for differences in behaviour and exposures outside of occupational environments due to the instigation (March 2020) and easing (June 2021) of the first UK national “lockdown” measures”.

Is there a typo here (June 2020?). ‘Covered the period’ is ambiguous, as it could refer to when the questionnaires were completed, or the time period referred to in the questionnaire. Suggest something along the lines of ‘Questionnaire invites were sent between October and November 2020 and questions within them related to participants’ recalled behaviour during two periods - March-May 2020 and June-July 2020’. Was the time of serological sampling in June 2020 only or was it done over 4 months (Mar 20 to Jun 20?). The wording is ambiguous.

Elsewhere it is written that “HCWs were sent questionnaires spanning Mar-Aug 2021.” I am struggling to reconcile this with other dates reported.

A flow chart to illustrate the chronology of serology sampling, consent, questionnaire issue / completion etc would help to clarify the methods.

These have been clarified and a flow chart has been added (Figure 1).

5. I was interested to see that BMI did not associate with risk of seropositivity – a contrast to findings from others e.g. PMID 35189888. Could the authors speculate why this may have been the case?

BMI was not assessed in this analysis. We assessed a univariate relationship with participants previously being told they were “overweight” or “obese” in a medical setting. This did not reveal any positive association, likely due to a combination of variability in measurements and reporting, plus lack of sensitivity of a binary variable when compared to discrete BMI calculations.

6. Table 1 (Participant characteristics) details age, sex, ethnicity and ‘COVID working’ – but there are lot more characteristics of interest here – job role, other exposures etc. It would be useful to flesh this out, esp with details of how different job roles were distributed.

This has been updated. All other demographic information and distribution of questionnaire responses is available in **Supplementary Table 1**. This has also been clarified in the main body of text.

7. What was the basis for the sample size? Was a power calculation done?

Minor

The was no indication for a sample size calculation, as this questionnaire was sent to all available respondents enrolled in the original longitudinal study. Additionally, there were no prior data available to assess differences between groups on the metrics assessed in the analysis to inform a sample size calculation.

This has been clarified in the text.

1. Typos:

Table C (p46/59): Pharmacy staff 7.51 – should this be 7/51?

Yes, this has been changed.

environments due to the instigation (March 2020) and easing (June 2021) of the first UK national “lockdown” measures’ – should this be June 2020?

Yes, this has been changed.

Elsewhere it is written that “HCWs were sent questionnaires spanning Mar-Aug 2021.’ Should this be March to July 2020?

Yes, this has been changed.

2. Abstract and elsewhere – if odds ratios adjusted, then they should be reported as such

This has been checked and updated where necessary.

VERSION 2 – REVIEW

REVIEWER	Chang, Chee-Tao Hospital Raja Permaisuri Bainun, Clinical Research Centre
REVIEW RETURNED	28-Sep-2022

GENERAL COMMENTS	The manuscript was adequately revised with improved clarity. I have no further comment, and I wish the authors all the best.
--

REVIEWER	Martineau, Adrian Queen Mary University of London
REVIEW RETURNED	22-Sep-2022

GENERAL COMMENTS	Thanks for comprehensive response to comments.
--